# Reproducibility Challenge: Reformer

**Artashes Arutiunian**[*]
aruthart@gmail.com

**Morgan McGuire**[*]
morg@hey.com

**Hallvar Gisnås**[*]
Norwegian Defence Research Establishment
hallvar.gisnas@ffi.no

**Sheik Mohamed Imran**
imrandude@gmail.com

**Dean Pleban**
DAGsHub
dean@dagshub.com

**Priyank Negi**
peiyank99@gmail.com

**David Arnoldo Ortiz Lozano**
tyoc213@tyoc.mx

## Reproducibility Summary

We attempt to reproduce the central claims of ICLR 2020 Paper "Reformer: The Efficient Transformer" (Kitaev et al. [2020]); that the techniques introduced enable performance on par with a traditional Transformer model while being much more memory-efficient and much faster on long sequences. This community effort reproduced claims around speed for long sequences and observed a reduction in memory usage. We could not match the performance of a traditional Transformer with Reformer. Finally, substantial coding effort was required, a lack of implementation documentation compounded this. We provide all code documentation in our GitHub[1].

**Scope of reproducibility**    The scope of this work is to verify the claims of memory efficiency and speed on longer sequences of the Reformer. We replicated only the NLP experiments due to limited computational resources.

**Methodology**    We first reimplemented the original Transformer model (Vaswani et al. [2017]) and which we then modified. We referred to the authors' code for the model and data pipeline. We used the fastai library (Howard and Gugger [2020]) for training, Weights and Biases (Biewald [2020]) for experiment tracking and nbdev (Howard [2019]) for development. All experiments were done in a single GPU setting.

**Results**    Claims around speed on longer sequences and reduced memory footprint were validated; as sequence length increased, Locality Sensitive Hashing ("LSH") Attention became faster and increasing the number of hashes improved performance. We could not achieve the performance of a traditional Transformer with Reformer. Some experiments were not run for as long as in the paper due to a lack of computational resources. Potentially the under-performance of our Reformer may be due to under-training, implementation differences or nuances in JAX vs Pytorch. Also, exploding gradients were encountered with mixed precision training and several model settings were found to be unstable depending on the random seed or learning rate.

**What was easy**    Obtaining the data was straightforward as they are commonly used benchmarks. There were no issues reproducing the data pipeline or Chunked Feed Forward layers and code for the Axial Positional Encodings was imported. [2]

**What was difficult**    Substantial effort was made to ensure a correct reimplementation. It was challenging due to many engineering design decisions or hyperparameters not being fully documented. Significant hyperparameter tuning was also needed.

**Communication with original authors**    The authors were receptive to email correspondence and clarified a number of implementation details.

---

[*]Corresponding author

[1]https://anonymous.4open.science/r/93b6a01d-e401-4344-93d9-57611c042a93/

[2]https://github.com/lucidrains/axial-positional-embedding

33rd Conference on Neural Information Processing Systems (NeurIPS 2020), Vancouver, Canada.

# 1 Introduction

One of the main characteristics of the Transformer (Vaswani et al. [2017]) is a large number of parameters, and costly training (see e.g. Brown et al. [2020]). Finding ways of matching the performance of the Transformer, but reducing the cost of training is, therefore, a democratizing effort. The Reformer (Kitaev et al. [2020], referred to later as the Reformer paper) attempts to achieve this by reducing the memory footprint of the Transformer and thus enabling modelling of deeper models and longer sequences within a fixed resource budget.

# 2 Scope of Reproducibility

The Reformer introduces several techniques for reducing the memory footprint of the Transformer while retaining acceptable performance. The Reformer paper claims that the Reformer "performs on par with Transformer models while being much more memory-efficient and much faster on long sequences". We break down this claim into the following sub-claims, based on the experiments in the paper:

1. Proof of concept: A Transformer with LSH attention can solve a sequence copy task that requires non-local attention lookups.
2. Attention approximation: The performance of the Reformer increases with the number of hashing rounds and is close to full attention performance at 8 hashes.
3. Scaling: Deep Reformer models: a) can be trained on very long sequences using a single accelerator (GPU or TPU core) and b) adding layers improves performance.
4. The performance of Reversible Transformers are similar to the Transformer on language modelling tasks.
5. Shared queries and keys in the attention module has minimal effect on performance in a Transformer.
6. Evaluation speed: The evaluation speed of LSH attention grows with the number of hashes but not with increased sequence length if the number of tokens is kept constant.
7. The Reformer reduces the memory footprint compared to the transformer.

We provide all code in an anonymous github repository. We have built an additional documentation web page of how experiments were performed. The documentation will be made available after the review period has ended. The notebooks which the documentation is based on is available in the /nbs folder of the repo. Finally, a pypi package with the reformer implementation will also be released.

# 3 Methodology

## 3.1 Reformer Techniques

The Reformer paper introduces five techniques which can be implemented separately or combined together into a full Reformer model:

1. **LSH attention** is an approximation of full attention that is more memory and time efficient for longer sequences.
2. **Shared keys and queries** in the attention module are introduced to make LSH work better by ensuring that the number of queries is equal to the number of keys within each bucket.
3. **Reversible Residual Layers** were first presented in Gomez et al. [2017]. Adding Reversible Residual Layers enables us to perform backpropagation without storing the activations in memory.
4. **Chunked feed-forward layers.** Processing large feed forward layers in smaller chunks.
5. **Axial positional encodings** are a more memory efficient way of encoding positions than standard learnable positional encodings.

For a more in-depth explanation of these techniques refer to our GitHub repository.

## 3.2 Datasets used

1. **Synthetic dataset**: A sequence of integers of the form 0w0w, where w is a sequence of integers from 1 to 128 of some length.
2. **enwik8** is a dump of English language Wikipedia from 2006 (Mahoney [2006]).

3. **WMT 2014 English-German** is a dataset of 4.5 million English and German sentence pairs (Bojar et al. [2014]). Downloaded from Huggingface.[3]

4. **Tiny Shakespeare** is a subselection of Shakespeare's works, downloaded from Huggingface.[3]

For all datasets, our code and documentation details the data pipeline.

### 3.3 Experimental setup and code

Our code and documentation with supplementary material such as full experiment notebooks are available on Github.[1] All experiments are logged to Weights & Biases. All experimentation was carried out with single GPUs, unlike the Reformer paper which used 8 x V100s. The total number of GPU hours was approximately 1 600 hours.

## 4 Results

We attempt to verify each of the claims stated in section 2.

### 4.1 Language modelling with enwik8 - experiment settings

For all enwik8 language modelling experiments we train a 3-layer Transformer language model ("TransformerLM") with a model dimension of 1024 and axial positional encoding, modified with the relevant Reformer technique, on the enwik8 dataset with a sequence length of 4096, unless noted otherwise. The Adafactor optimizer and a batch size of 8 was used, via Gradient Accumulation. Experiments were run on 15GB and 12GB GPUs and training was carried out in full-precision. Experiment configuration is detailed in our documentation.

### 4.2 LSH attention analysis on synthetic task

The synthetic task is a sequence copy task. The input sequences are of the form 0w0w, where w is $n$ random integers between 1 and 128. We use a language model to predict the next token in such sequences. The learning objective is to copy the final half of a sequence from the first half. See our documentation for details on the task.

**Claim:** A full attention Transformer can solve this task perfectly. An LSH-attention Transformer can also solve it, but with decreasing performance as the number of hashing rounds decrease. A model trained with one type of attention can also be evaluated with a different type of attention.

**Experiments** For this task we train a shared-QK TransformerLM and three LSH Language Models ("LSH-LM") with 1,2 and 4 hashing rounds respectively. We trained for 150 000 steps with a batch size of 64, a training size of 12 800, and a validation size of 1 280. We used hyperparameters as described in the paper, and other defaults suggested in the Trax Github repository.[4] See our SyntheticConfig class for experiment defaults. We evaluated the models with full attention and LSH attention with 8,4,2 and 1 hashing rounds respectively.

**Results** Results from a single run are summarized in table 1. The first column shows how the model was trained, the subsequent ones how it was evaluated. See our documentation for analysis setup.

Table 1: Summary of accuracy on the synthetic task (percentage).

| Training | Evaluation | | | | |
|---|---|---|---|---|---|
| | Full Attention | LSH-8 | LSH-4 | LSH-2 | LSH-1 |
| Full Attention | 100.0 | 1.4 | 1.9 | 3.0 | 4.6 |
| LSH-4 | 46.5 | 99.7 | 99.7 | 93.05 | 77.6 |
| LSH-2 | 75.9 | 96.6 | 97.5 | 97.1 | 86.1 |
| LSH-1 | 70.7 | 76.6 | 79.7 | 79.3 | 56.1 |

We can partly support the claim, but there are three clear differences when we compare our results to Table 2 of the Reformer paper. First of all, our results are worse than the Reformer paper. However appendix A also shows that models using shared-QK attention have large variation in training loss. This suggests results are somewhat random, and depend on the specific training setup. Secondly, models trained with LSH attention and validated with standard attention do **much better** in our experiments. And thirdly, the model trained with standard attention and validated with LSH attention does **much worse** in our experiment. We have been unable to explain these differences.

---

[3]https://huggingface.co/datasets/
[4]https://github.com/google/trax

### 4.3 Effect of sharing QK

Shared Query-Key ("shared-QK") Attention was used in the Reformer paper as it is needed for LSH. It also helps to further reduce the memory footprint of the model.

**Claim:** A shared query-key space does not perform worse than regular attention

**Experiments**    To investigate this we train a standard TransformerLM and a TransformerLM with shared-QK attention on the enwik8 dataset. The Bits-Per-Character (BPC) metric was used.[5] They were trained for 10 epochs and the mean validation BPC of 3 training rounds was taken, results are shown in figure 1.

**Results**    After experimentation, we cannot validate the claim that shared-QK attention does not perform worse than standard attention for this experimental setting. Nor did we see the effect of shared-QK attention training slightly faster as noted in the Reformer paper. Potentially with additional training shared-QK attention performance will converge with standard attention, we leave this to future work.

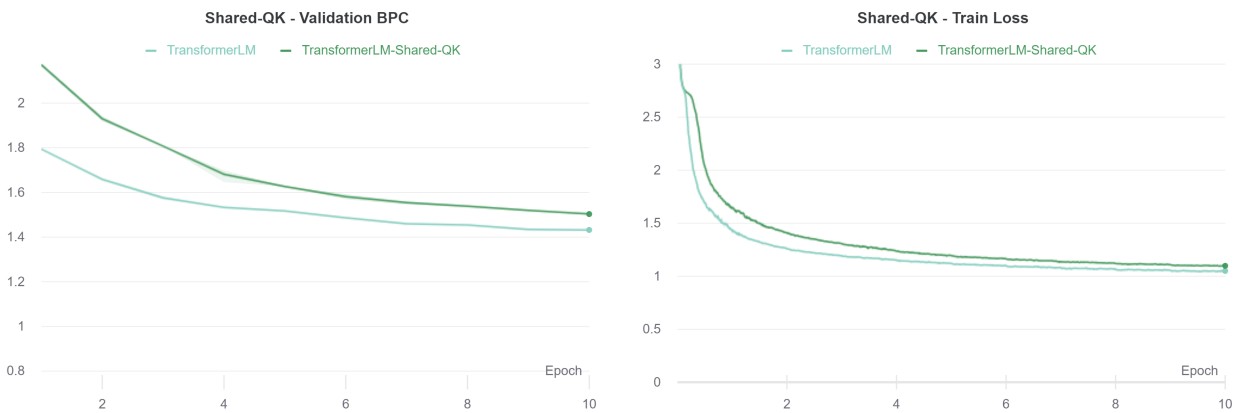

Figure 1: Validation BPC and training loss for TransformerLM and Shared-QK TransformerLM

### 4.4 Effect of reversible layers

**Claim:** Reversible Residual Layers in a Transformer enable more memory efficient training and do not come at the expense of model accuracy

To validate this claim we train a TransformerLM and full sequence-to-sequence Transformer ("ReversibleTransformer") using Reversible Residual Layers and compared them to their standard Transformer equivalents.

**Reversible Language Model on enwik8 experiment**    To investigate this claim we trained on sequences of length 4096, since training this model ("ReversibleLM") on full 64k sequences from enwik8 as per the Reformer paper was not feasible given our computational budget. Figure 2 shows the mean of 3 runs for each model-type.

**Results**    We could not validate the claim that Reversible Residual Layers do not have a significant impact on language model performance due to the sizeable difference of 0.11 mean validation BPC between the ReversibleLM and the baseline TransformerLM. Potentially this difference can be reduced with further training, we will leave this to further work.

### 4.5 Reversible Transformer on translation task experiment

We train a full Reversible Transformer, a 12-layer sequence-to-sequence Transformer model with Reversible Residual layers on the WMT-14 en-de translation dataset. Given the short sequences in this dataset, a sequence length of 256 was used with a batch size of 64 and the AdamW optimizer. Gradient Accumulation was used when training with a 12GB GPU. Training was carried out for 2 epochs for the Reversible Transformer and 1 epoch for the standard Transformer. Full-precision was used in both cases. The one-cycle learning rate schedule from fast.ai was used, with a maximum learning rate of 1e-4, initial div of 5, and a percent start of 12.5%

---

[5]The Reformer paper used the BPD, which is equivalent to BPC when evaluating characters.

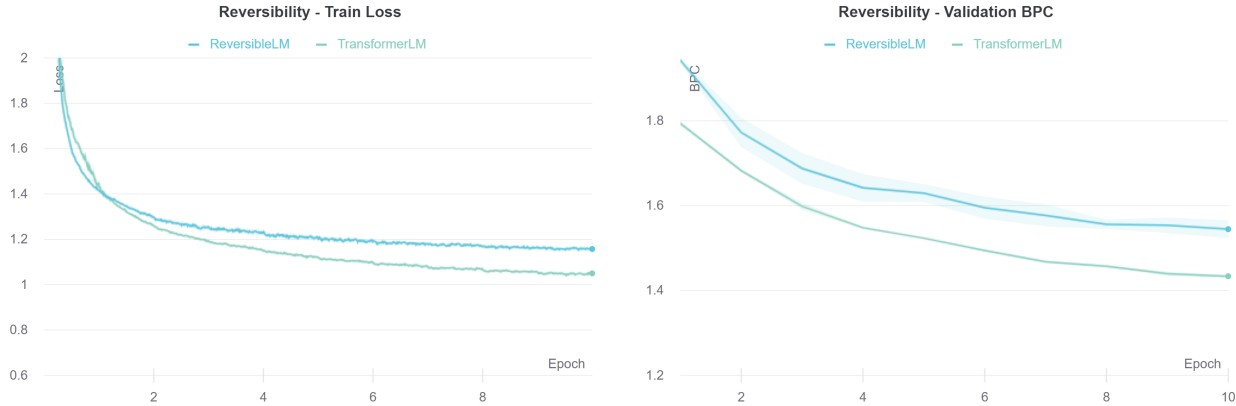

Figure 2: Training loss and validation BPC for ReversibleLM and TransformerLM on enwik8

**Results**    From our experiments, see figure 3, we could not not validate the claim that Reversible Residual Layers do not have a significant impact on Transformer model performance. Our validation CorpusBLEU score for a standard Transformer trained for 1 epoch outperformed a ReversibleTransformer that was trained for 2 epochs.

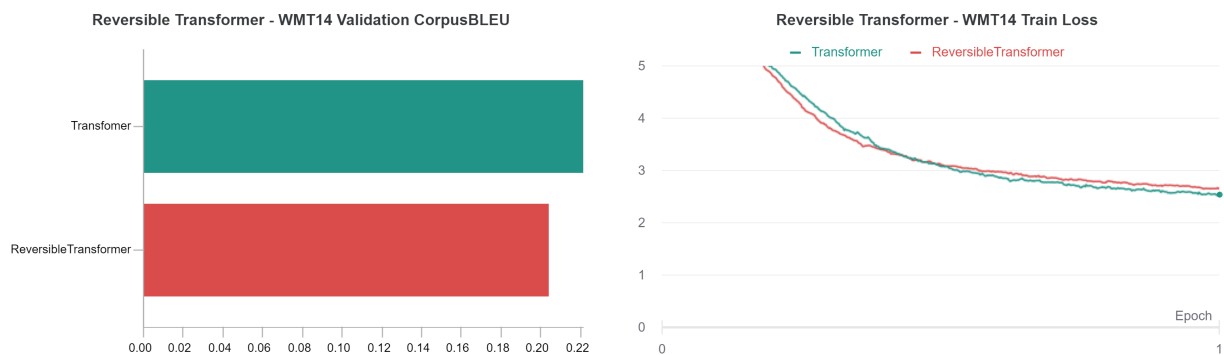

Figure 3: Performance on WMT-14 en-de translation dataset for ReversibleLM and Transformer.

### 4.6    Effect of number of hashing rounds on the performance

When using LSH for chunking there is a small probability that similar tokens will end up in different buckets, and therefore attention will not be computed for these tokens. To reduce this probability we can do multiple rounds of hashing.

**Claim:** Performance of Transformer with LSH attention increases with the number of hashing rounds and is close to full attention performance at `n_hashes = 8`.

**Experiment**    To validate this claim we train and compare models with different numbers of hashing rounds on the enwik8 dataset. We train a LSH-LM and a stanard TransformerLM for 10 epochs.

**Results**    Figure 4 shows training and validation loss for LSH-LM runs with a number of hashing rounds from 2 to 16 compared to the TransformerLM. Losses for one run of each setting are reported. Our results verify the claim and illustrate that training and validation losses tend to improve with an increasing number of hashing rounds. To be noted that both time and memory requirements also increase with `n_hashes`.

Also while training losses are very close for larger `n_hashes` and full attention, models using LSH seem to have slightly higher generalization error.

### 4.7    LSH attention evaluation speed

**Claim:** The evaluation time for LSH attention increases with the number of hashes but not with increased sequence length when the total numbers of tokens are kept constant.

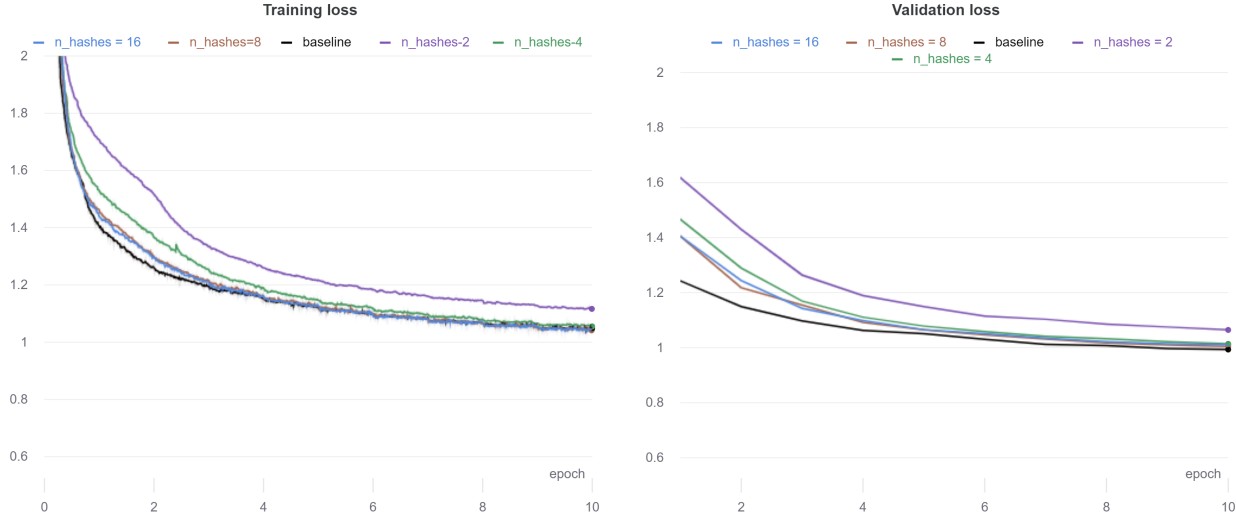

Figure 4: Training and validation loss as a function of number of hashes for TransformerLM with LSH attention

**Experiment**   To verify the claim we test evaluation time of the LSH-LM on the synthetic task with hyperparameters as indicated in the right part of Figure 5 of the Reformer paper. We were unable to complete the longest sequence lengths for full attention due to memory limitations on a single GPU.

**Results**   Our results support the claim and are summarized in figure 5.

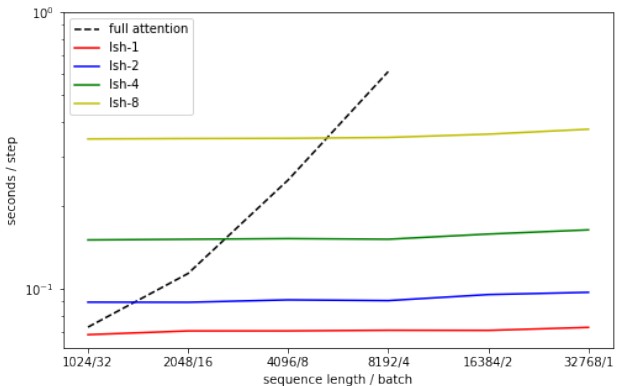

Figure 5: Evaluation speed as a function of number of hashes and sequence length

## 4.8   Deep Reformer models

**Claims:** 1) Deep Reformer models can be trained on very long sequences using single accelerator (GPU or TPU core); 2) Adding layers to Reformer models improves performance.

**Experiment**   To test these claims we train full ReformerLMs of increasing depth on sequences of length 16 384. We use the enwik8 dataset. Our experiments are designed to run on a single 12GB GPU. We trained models with 3, 6 and 12 layers for 4 epochs, training deeper models was not possible with our computational budget.

**Results**   We confirm claim (1), deeper models can be trained on a single accelerator as expected, given the fact that Reversible layers have $O(1)$ memory complexity. For claim (2), our results in figure 6 are indicative of better performance for deeper models, but longer training runs are needed to confirm this.

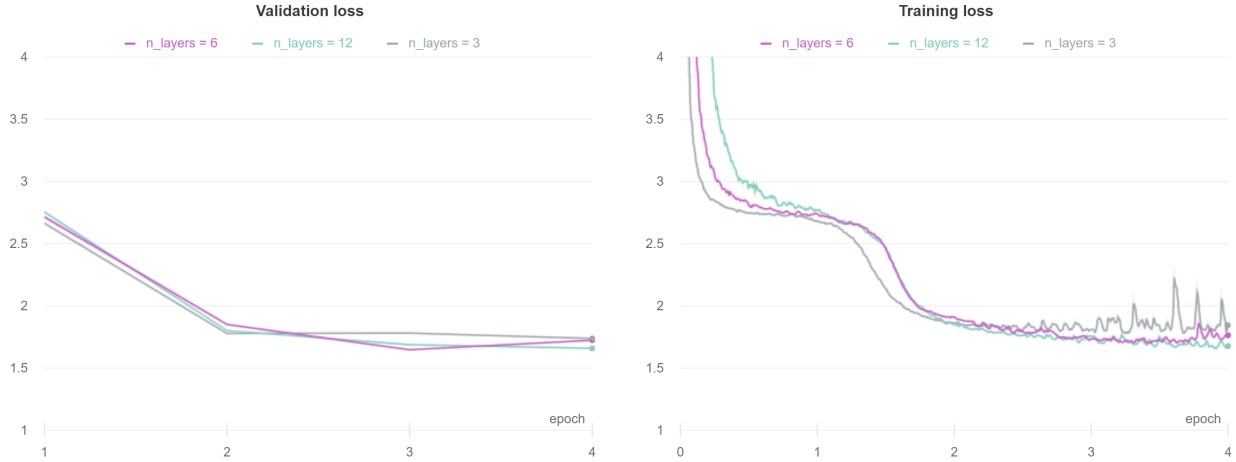

Figure 6: ReformerLM validation and training loss as a function of number of layers on enwik8

## 4.9 Memory Consumption

A central motivation of the Reformer paper is that it has a reduced memory footprint as compared to a standard Transformer.

**Claim:** The Reformer reduces the memory footprint compared to the Transformer. The claim is summarized in Table 5 of the Reformer paper.

**Experiments** To investigate the claim we log the memory allocation of various reformer combinations for one training step with the tiny Shakespeare dataset. The sequence length for all experiments was set to 4096, with a batch size of 1. We investigate the following combinations: a) Comparing the TransformerLM, LSH-LM, ReversibleLM and full ReformerLM; b) Comparing ReformerLMs with different number of hashes.

**Comparing TransformerLM, LSH-LM, ReversibleLM and the full ReformerLM** Figure 7 (left) shows the memory usage for the models compared. We see that the Transformer stores activations for the forward pass during training, and that memory is gradually released as the backward pass is completed. In this case we have six distinct jumps in memory which correspond to its 6 layers. With the LSH-LM, memory allocation is mostly parallel to the transformer, but since the LSH attention computation is cheaper, the actual memory consumption is lower. For the ReversibleLM memory doesn't accumulate over the forward passes, since it recalculates intermediate activations for the backward pass. Note that we observe 6 peaks for the forward pass, each peak corresponds to full attention matrix computation. Another 6 higher peaks correspond to backward pass. The full ReformerLM includes both LSH attention and Reversible Residual Layers. Memory allocation therefore doesn't increase with number of layers, and since LSH attention is cheaper than standard attention, the peak memory usage is smaller than for the ReversibleLM.

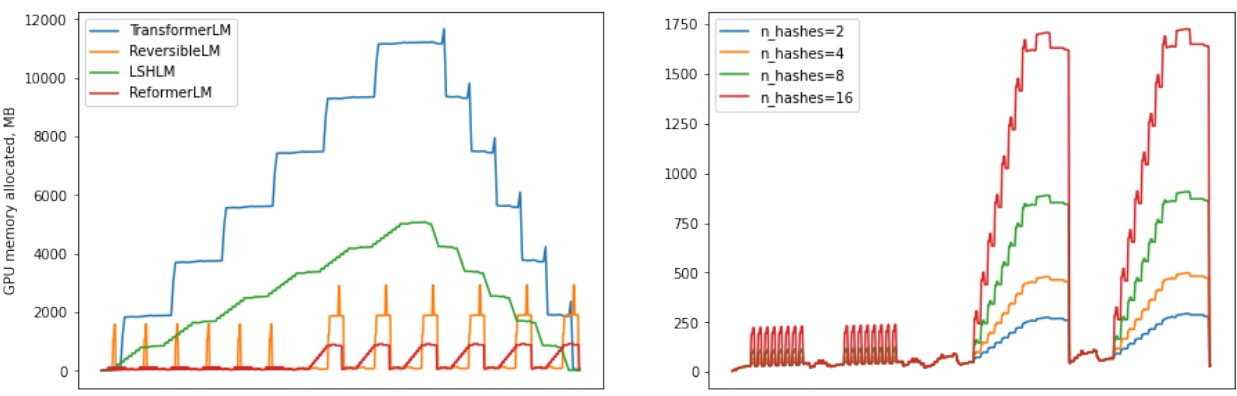

Figure 7: (left) Memory allocation of TransformerLM and Reformer variants during forward and backward pass; (right) Memory allocation as a function of number of hashes for ReformerLM

**Reformer Memory vs Number of Hashes**   A 2-layer ReformerLM with a sequence length of 4096 was used for this experiment. In section 4.6 we see that increasing the number of hashes leads to better performance. But since we store the result of each hashing round we expect memory to grow linearly with the number of hashes. Note that it's only during the LSH attention calculation that the number of hashes matter. I.e. the intermediate shape of LSH attention is `[batch_size, n_chunks, chunk_size, chunk_size*2]`, where n_chunks is the product of the number of hash buckets per round and the number of hash rounds, `n_chunks = n_buckets * n_hashes`. The output of LSH attention is `[bs, seq_len, d_model]` and is independent of the number of hashes. Figure 7 (right) confirms that peak memory scales linearly with number of hashes. The memory peak happens at the backward pass during calculation of LSH attention. Appendix B shows a similar experiment that investigates how the number of layers affect memory consumption.

**Results**   Our experiments have verified that the Reformer has a much smaller peak memory allocation as compared to the Transformer. We have also shown that the reformer can scale to much deeper models than the Transformer within a fixed budget. The main bottleneck of the reformer w.r.t. memory is the number of hash rounds used.

# 5   Discussion and Conclusion

We have validated a number of claims such as superior evaluation speed for longer sequence lengths, how LSH attention is close to approximating full attention, that scaling deeper Reformer models works on a single GPU and that Reformer has a reduced memory footprint compared to the Transformer.

However we cannot validate the claim that performance of a Transformer model with either Reversible Residual Layers or Shared-QK attention performs on par with a traditional Transformer. There may be a number of explanations, our lack of compute compared to the original authors being the most likely factor.

This reproducibility work involved a non-trivial amount of engineering to ensure our reimplementation was as faithful to the original paper as we could make it. The Reformer paper introduced five new techniques meaning that substantial effort had to be made to ensure a correct reimplementation of each. LSH attention is, for example, quite complex compared to normal dot-product attention. While there were Reformer code resources available, many design decisions or hyperparameters were not fully documented in the paper.

Several of the experiments are also quite computationally intensive for a single GPU setup, taking over 30 hours per epoch. The enwik8 64K sequence length and imagenet-64 experiments could not be tested for this reason. Nevertheless the use of shorter sequence lengths enabled us to thoroughly investigate the claims made.

In our experimentation we observe that Reformer enables training of deep models with long sequences on a single GPU, supporting the democratization of machine learning. Practitioners interested in using Reformer should note however that we also observed a non-negligible increase in time complexity due to the trade-off of memory for compute in the Reversible Residual Layers and the trade-off of time for increased accuracy as the number of hashes used increases. This behaviour is also observed elsewhere, e.g. Katharopoulos et al. [2020] and Tay et al. [2020].

As this was a community effort with many contributors we were forced to improve our development processes during the project. We started using nbdev (Howard [2019]), an open-source dev-ops system with documentation, testing and full CI/CD integrated into GitHub. This resulted in a much smoother development process for the team.

All of our code and documentation is available on GitHub[1]. Of particular interest to future practitioners might be our integration of the ReformerLM and Transformer sequence-to-sequence training into the fastai training loop, as well as the memory profiling scripts used in our memory consumption experiment.

# 6   Acknowledgments

There are a number of contributions that we would like to acknowledge; Weights & Biases for supporting this community work by providing a team account, DAGsHub (Pleban and Smoilovsky [2019]) for providing computational resources for experimentation and Phil Wang (@lucidrains on GitHub[6]) for his Pytorch Reformer implementation as well as a number of other transformer implementations which made this complex implementation much smoother. Finally, we would like to acknowledge that this work originates in the fastai community and to thank Jeremy Howard, Sylvain Gugger and all of the contributors to the library and the forums for continuing to make deep learning more accessible, more friendly and more performant.

---

[6]`https://github.com/lucidrains`

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

# A  Training dynamics

**Supplementary Shared-QK Experiment**    During the experiment phase of the synthetic task (section 4.2), we used a TransformerLM with Shared Query-Keys. We expected this model to solve the relatively simple synthetic task to perfection, but we found it difficult to train the model to convergence even when training for 750 epochs. When we instead used a standard TransformerLM (i.e. with separate Queries and keys), the model consistently converges in 7-8 epochs.

Figure 8 illustrates several runs on identical setup, changing only the model seed. Out of five runs with shared query-key only one model converged after about 200 epochs. For our standard TransformerLM the three runs converge within 7-8 epochs.

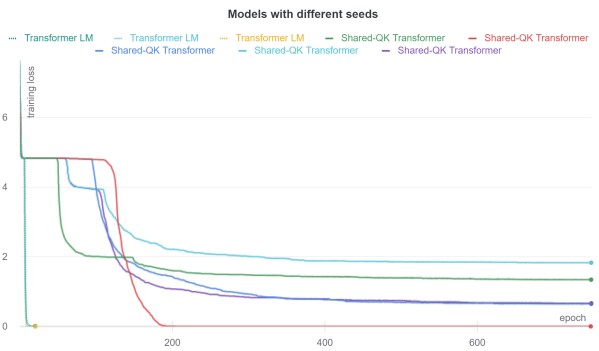

Figure 8: Training loss for various seeds and models on the synthetic task

# B  Reformer vs Number of Layers

In this experiment we compare ReformerLMs with `n_layers = 2,4,6,8,10,12`. All models use 8 hashes. The results in figure 9 show a slight increase in memory as the number of layers grow. The Reformer isn't accumulating memory during forward and backward passes, but additional memory is used because more layers increases the number of model parameters.

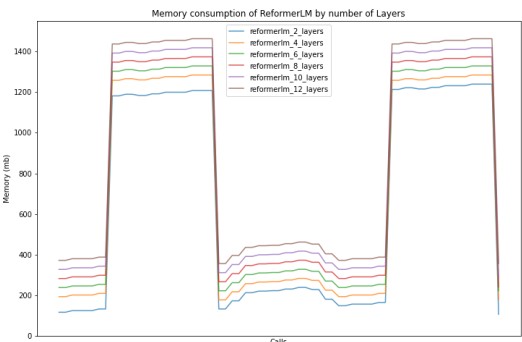

Figure 9: Memory allocation as a function of number of layers for ReformerLM

