# OpenReview forum: "Reproducibility Challenge: Reformer"
_ML_Reproducibility_Challenge/2020 — Reject_

### Official Review · AnonReviewer2 · 2021-03-01
**A report with compressive experiments**

**Rating:** 8
**Confidence:** 5

**Review:**

This study attempted to verify the claims of the Reformer paper:
1. LSH based attention achieves higher speed for long sequences: reproduced
2. Reformer has low memory footprint than baseline: reproduced
3. Reversible layers and shared key queries achieve similar performance with baseline: not reproduced
4. LSH based attention achieves similar performance as full attention: reproduced

The authors reported both training and validation losses for 1, 3 and 4. Claim 2 was supported by Figure 7.

The study also pointed out that the reformer may increase training/inference time due to increased rounds of hashing to match the baseline performance.


**Familiar With The Original Paper:**

I have read the original paper

**Reproducibility Summary:**

Report has summary

---

### Official Review · AnonReviewer1 · 2021-03-02
**A very difficult reproducibility task well performed**

**Rating:** 8
**Confidence:** 4

**Review:**

General Remarks

The fact that the authors failed to anonymize the report by mentioning their names and also by referencing a public Github repo.  That had no effect on reviewing their work, so I don't consider it a problem. I think they did overall a good work reproducing the result and providing a practical ablation study, given the scale of the experiments. The authors did follow the template and included the summary page correctly.

Positives

The paper breaks down the scope of the report into verifying 7 claims of the paper. I also liked the glossary that they introduced in 3.1. Overall the authors managed to reproduce many claims. Some claims were not reproduced, because of several engineering technicalities that the original paper introduced.

This was an insightful observation

"In our experimentation we observe that Reformer enables training of deep models with long sequences on a single GPU,
supporting the democratization of machine learning. Practitioners interested in using Reformer should note however
that we also observed a non-negligible increase in time complexity due to the trade-off of memory for compute in the
Reversible Residual Layers and the trade-off of time for increased accuracy as the number of hashes used increases.
This behaviour is also observed elsewhere, e.g. Katharopoulos et al. [2020] and Tay et al. [2020]. "

Need to be fixed

There are too many footnotes on websites. They should be moved to the references section. It will make reading easier. In fact, it might be more appropriate to create an appendix since the websites contain a lot of information that is important for understanding the report.
The mini Ablation study on how to reproduce table 1 is very useful and points a weakness of the paper.

The 4.2 section is confusing and needs to be rewritten. The authors make a reference to Github. This sentence "The first half of the sequence is masked from the loss function, so the goal for the model is to learn that midway through the sequence it has to repeat 0w" doesn't make sense to me.

Table 1 the column names need to be top-aligned




**Familiar With The Original Paper:**

I have read the original paper

**Reproducibility Summary:**

Report has summary

---

### Decision · Program_Chairs · 2021-03-31

**Decision:**

Reject

**Comment:**

While the technical reproducibility is well carried out, the report itself lacks proper formatting and readibility for its results to be easily interpreted by readers.